# SiFT: uncovering hidden biological processes by probabilistic filtering of single-cell data

Zoe Piran [1] & Mor Nitzan [1,2,3]

Cellular populations simultaneously encode multiple biological attributes, including spatial configuration, temporal trajectories, and cell-cell interactions. Some of these signals may be overshadowed by others and harder to recover, despite the great progress made to computationally reconstruct biological processes from single-cell data. To address this, we present SiFT, a kernel-based projection method for filtering biological signals in single-cell data, thus uncovering underlying biological processes. SiFT applies to a wide range of tasks, from the removal of unwanted variation in the data to revealing hidden biological structures. We demonstrate how SiFT enhances the liver circadian signal by filtering spatial zonation, recovers regenerative cell subpopulations in spatially-resolved liver data, and exposes COVID-19 disease-related cells, pathways, and dynamics by filtering healthy reference signals. SiFT performs the correction at the gene expression level, can scale to large datasets, and compares favorably to state-of-the-art methods.

Cells encode information in their gene expression profiles about different facets of their identity, such as their spatial location within tissues, cell cycle phase, and disease stage. Recent years have seen a surge in computational methods for the reconstruction of such cellular facets from single-cell RNA-sequencing (scRNA-seq) data[1]. While these reconstruction methods were proven successful for the recovery of diverse signals, including spatial[2–4] and temporal[5,6] signals, most of these methods focus on reconstructing a single signal in the data, relying either on its dominance or based on sufficient prior knowledge (such as known marker genes). However, since cells encode multiple signals about their intrinsic state and extrinsic environment, focusing on a single signal (measured or recovered) is insufficient and may overlook key cellular attributes. For example, while the cells' spatial organization may be the dominant signal in a single-cell dataset, it may overshadow a temporal regulation pattern. In such a scenario, information about the reconstructed (e.g., spatial) signal can be used to filter it from the data and reveal further complex hidden attributes. Another example involves case–control comparisons, where information about healthy controls can be filtered from cells sampled from patients along disease progression to uncover the subpopulation of cells affected by the disease and characterize their response.

The signal filtering approach was previously explored for removing unwanted sources of variation as a preprocessing step. For example, multiple computational methods have been proposed for data integration and the removal of batch effects (e.g., bbknn[7], Harmony[8], and scVI[9,10]), or specifically for removing the cell cycle signal, a major source of bias in single-cell data[11–15]. This bias introduces large within-cell-type gene expression heterogeneity that can obscure the differences between cell types. In turn, the latter can resurface once the cell cycle signal is filtered out. Yet, these methods are task-specific; data integration methods (apart from scVI) typically focus on a categorical factor to encode the different groups (batches) in the data, and cell cycle filtering approaches tend to account for known informative genes[13,14] or take advantage of the signal's cyclic structure[11]. Similarly, scPrisma, a spectral template-matching method, was recently proposed for the inference, filtering or enhancement of underlying signals based on prior knowledge of their structure (such as the cyclic structure of the cell cycle)[16]. In addition, most integration methods, including scVI, provide a correction at the level of a joint latent representation of the cells, and not for the original count matrix or for individual genes, thus limiting the applicability of existing downstream analysis tools and requiring the development of case-specific methods[17]. Hence, altogether, the

[1]School of Computer Science and Engineering, The Hebrew University, Jerusalem, Israel. [2]Racah Institute of Physics, The Hebrew University, Jerusalem, Israel. [3]Faculty of Medicine, The Hebrew University, Jerusalem, Israel. ✉e-mail: mor.nitzan@mail.huji.ac.il

above methods cannot be used to filter out a generic biological signal from single-cell data.

To this end, Satija et al.[18] suggested removing technical and cell cycle effects using a linear regression model. In this setting, independent linear models are fitted to predict gene expression with respect to a set of predefined variables. Then, for each variable independently, the fitted linear model is regressed from each gene. However, this strategy is restricted to the fit of the linear model and does not allow for additional user inputs to adjust the removal process to ensure that desired biological components are not removed from the data.

Here we introduce SiFT (Signal FilTering), a diverse and robust framework for filtering signals induced by different biological processes in single-cell data, thus uncovering underlying processes of interest. To do so, we compute a probabilistic cell–cell similarity kernel, which captures the similarity between cells according to the biological signal we wish to filter. Using this kernel, we obtain a projection of the cells onto the signal in gene expression space. By deducting this projection from the original data, we filter the signal-related information and uncover additional, hidden cellular attributes.

We begin by demonstrating that SiFT can successfully and efficiently remove sources of unwanted variation in the data while preserving biological attributes. This is showcased by applying SiFT to filter nuisance signals in *Drosophila* wing disc development single-cell data and removing cell cycle effects from a semi-synthetic single-cell dataset mimicking the existence of two sub-clones, where SiFT outperforms state-of-the-art methods. Next, we exemplify SiFT's ability to expose and enhance underlying biological signals. To do so we use prior knowledge regarding liver zonation to filter the spatial signal from single-cell liver data, thereby enhancing the temporal circadian signal encoded by the cells. In addition, we apply SiFT to spatially resolved data of liver regeneration where global subpopulations involved in the regeneration process are exposed by filtering the local neighborhood of cells using the corresponding spatial coordinates. Finally, we demonstrate the application of SiFT in a case–control setting in the context of COVID-19 progression. Healthy samples are used as a reference to filter the healthy signal and identify disease-related dynamics. SiFT is available as an open-source software package https://github.com/nitzanlab/sift-sc, along with documentation and tutorials at https://sift-sc.readthedocs.io.

## Results
### Revealing hidden biological signals using SiFT
The SiFT framework leverages known relationships between cells to expose additional, underlying structures in single-cell data (Fig. 1). Consider the scenario where each cell has two attributes, which we will term here shape and color. Now, assume that we have experimentally measured, or can computationally recover, the color identity (e.g., by coupling known marker genes to a clustering of the data). The biological signal that remains meaningful to uncover is the shape of the cell, for which no prior knowledge exists. By using SiFT to remove the color signal from the data we can uncover the shape signal (Fig. 1a).

SiFT takes as input both an expression count matrix, as well as knowledge about a specific signal encoded by attributes of the cells. The latter can be provided as a mapping of the cells using deterministic labels (e.g., cell cycle stage or spatial coordinates), a set of marker genes, pseudotime ordering, or a latent space representation of the cells ("Methods", Fig. 1b). These attributes are used to compute a cell–cell similarity kernel with respect to the encoded signal (Fig. 1c). Alternatively, the mapping to the signal can be based on a population of reference cells (e.g., control cells in a case–control setting). Then, the cell–cell similarity kernel is computed only using the reference cells to capture the reference (e.g., control) signal.

In general, the kernel captures distances between cells in the signal space, thus encoding the cells' similarity concerning the signal to be filtered. We define three main variants of cell–cell similarity

kernels: a mapping, k-nearest-neighbor (knn), and a distance kernel. The mapping kernel relies on a stochastic or deterministic mapping of the cells to a given domain. Such mapping results in cell labels, including cell-type labels or temporal labels generated by binning of a pseudotime trajectory. In such a case, the resultant kernel will follow a block structure, grouping cells associated to the same label. The knn and distance kernels rely on a joint space representation of the cells over which a corresponding distance metric can be defined. Such spaces can be generated, for example, by restricting the original single-cell data to a set of marker genes, or a latent space representation of the cells based on single-cell variational inference (scVI)[10]. Given such a representation, the knn kernel is defined as the row-normalized weighted adjacency matrix based on the distance measure used to identify each cell's neighbors. The distance kernel uses the cell distances directly, transformed to valid probabilities using the SoftMax function. The choice of the kernel type relies on the structure and knowledge regarding the existing signal attribute. The main difference is in the coarse representation captured by the mapping kernel in contrast to the weighted similarities obtained by the knn and distance kernels ("Methods", Fig. 1c). Alternatively, the user can supply a precomputed cell–cell similarity kernel that captures information they aim to filter.

Given a cell–cell similarity kernel, we obtain a projection of the expression count matrix onto the signal we seek to filter using matrix multiplication ("Methods"). With this construction, the projection can be interpreted as the portion of the expression associated with the known signal. Hence, we can deduct it from the original count matrix, and obtain a filtered representation of the data which represents the deviation of each cell's gene expression from the expected, or typical expression in its neighboring cells, to capture attributes associated with additional biological signals (Fig. 1d). Thereby, SiFT provides a corrected count matrix (which can contain negative values or be corrected by a pseudocount; "Methods"). Now, diverse analysis tools can be applied to study the filtered data and explore the underlying biological signals it encodes (Fig. 1e).

### SiFT efficiently removes unwanted sources of variation
Experimental data often contains unwanted sources of variation which obscure the biological signal of interest. These can be discrete (e.g., sex label) or continuous (e.g., cell cycle phase) signals. A desired preprocessing step is to remove such sources of variation. An optimal removal procedure should: efficiently remove the unwanted signal while preserving the biological attributes, be generic and adaptive to diverse settings, and be easily included in the analysis pipeline (in terms of implementation and scalability with respect to dataset size). We turn to show that SiFT meets all of these criteria.

As a start, we consider a single-cell transcriptomics dataset of the *Drosophila* wing disc, previously shown to suffer from unwanted sources of variation due to cell cycle and sex signals[19]. The original embedding of the cells reflects the bias induced by both the cell cycle and sex signals within each batch (Fig. 2a). Using SiFT, we can filter these signals and uncover the underlying biologically meaningful variability. We apply SiFT to filter both cell cycle and sex-related variation using a corresponding reported set of marker genes (Supplementary Table 1). Namely, the marker genes' expression is used to construct a knn kernel, capturing the neighborhood of the cells with respect to the cell cycle and sex signals (Supplementary Fig. 1). Then, this kernel is used to filter out the unwanted signals.

The SiFT-corrected embedding of the cells shows a homogeneous representation with respect to the sex and cell cycle phase signals, which we aimed to filter (Fig. 2a). That is, in contrast to the original data, labels are not visibly separable in the latent space representation (Fig. 2a), and the marker genes' spatial gradients are removed (Supplementary Fig. 2). This qualitative result is supported quantitatively by the graph iLISI score, evaluated for each biological batch

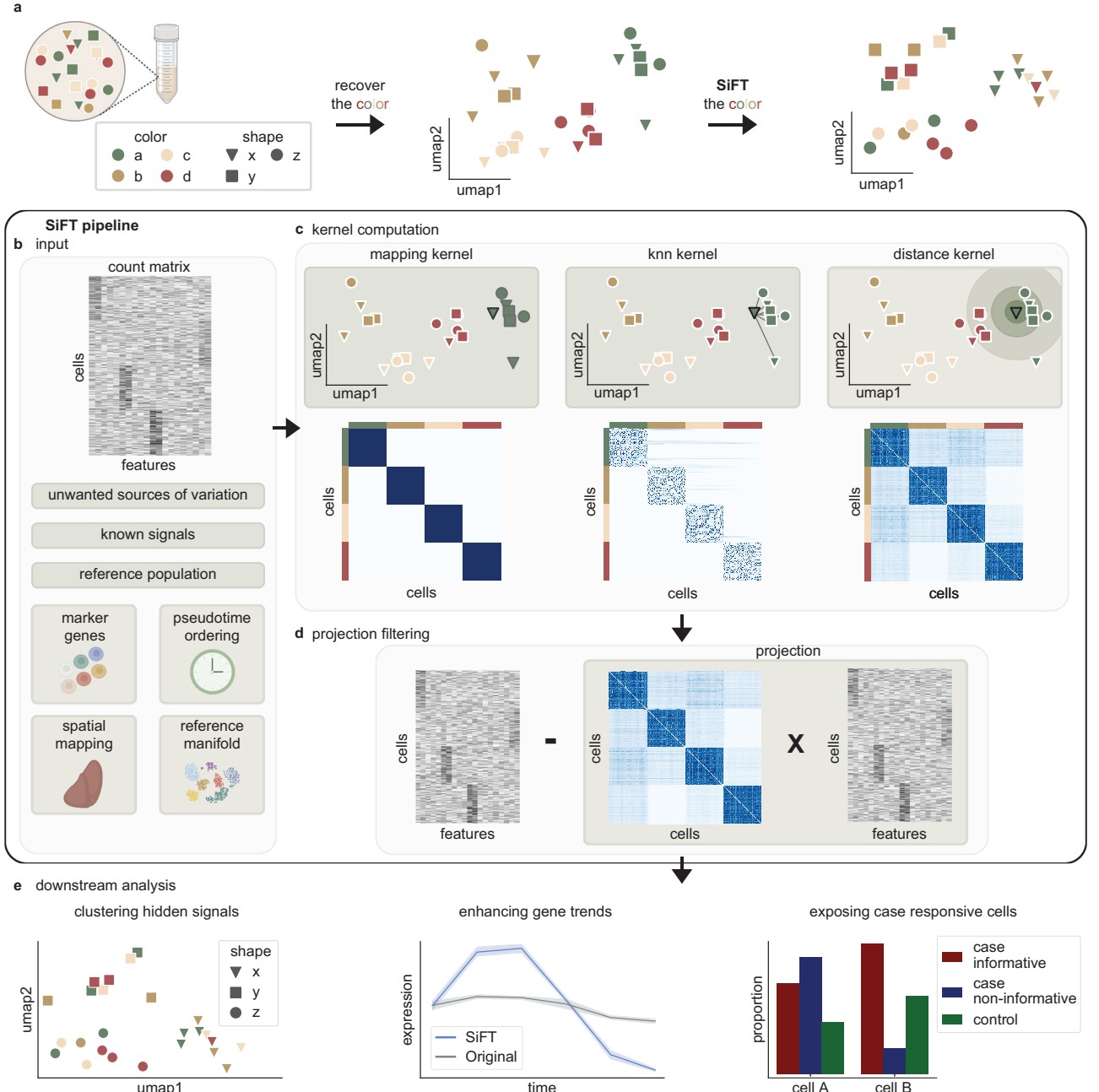

**Fig. 1 | Overview of the SiFT algorithm. a** Conceptual illustration of the application of SiFT to cells where each cell has two attributes, shape and color. Applying SiFT will expose the shape attribute, based on our existing knowledge of the color structure. Figure is created with BioRender.com. **b**–**d** The SiFT pipeline. **b** input; SiFT takes as input a count matrix and a pre-computed mapping of the cells. The mapping can be either continuous or discrete and either univariate or multi-dimensional. Figure is created with BioRender.com. **c** Cell–cell similarity kernel computation; the cell–cell similarity kernel is computed based on the given signal mapping. Columns correspond to the different kernels, the mapping kernel (left), the knn kernel (center), and the distance kernel (right). The top row illustrates the cell similarity structure captured by the kernel and the bottom row illustrates the cell–cell similarity kernel. Figure is created with BioRender.com. **d** Projection filtering; filtering is performed by projecting the kernel onto the count matrix and deducing the projection from the original count matrix. **e** Downstream analysis; after filtering, the hidden structure is exposed and easily recovered in downstream analysis. SiFT allows labeling according to hidden signals (left), enhances underlying gene trends (middle), and uses a reference control population to identify cells that are responsive for the case, correcting the naive abundance test and identifying that cells of type B contain a larger population of cells informative of disease state (right).

independently, with respect to cell cycle labels as well as joint cell cycle and sex labels ("Methods"). This quantitative evaluation shows that SiFT can successfully remove unwanted variation in single-cell data (Fig. 2b). Further, SiFT is robust to kernel as well as hyper-parameter choice; specifically, the performance is robust to different values of $k$ (the number of neighbors for the knn kernel) and to kernel choice (shown for the distance kernel using the same input as the knn kernel

above, Supplementary Fig. 3). However, using a mapping kernel, which takes as input coarse labels, reduces the performance. This is expected, as the mapping kernel induces a block structure that gives uniform weight to all cells of the same label without further tuning for local neighborhoods of the individual cells, in contrast to the knn kernel.

Furthermore, SiFT compares favorably to available baselines for this task, including Scanpy's regress out function (a Python

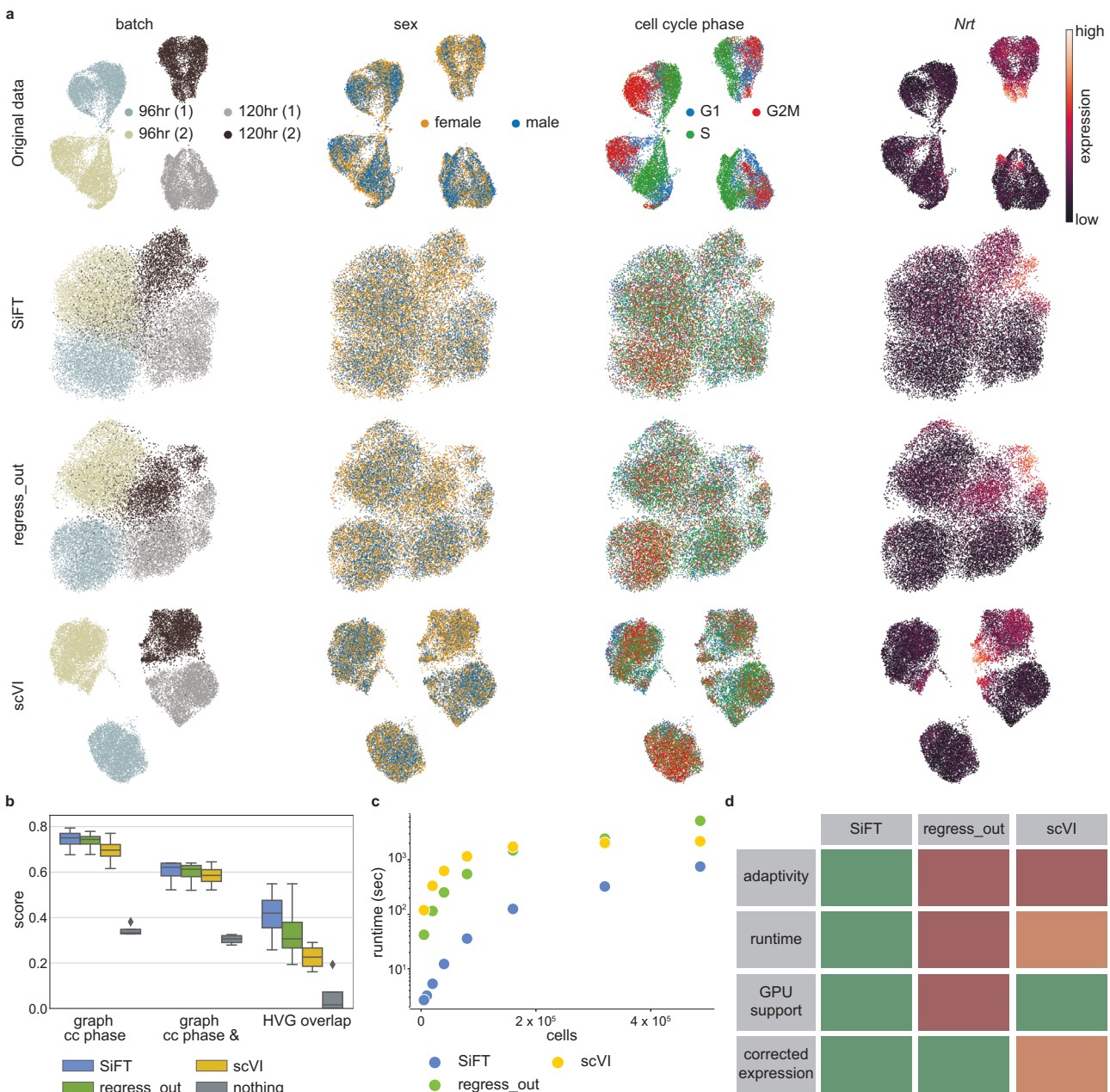

**Fig. 2 | SiFT removes unwanted variation from single-cell data. a, b** Removal of unwanted sources of variation from transcriptomics of the *Drosophila* wing disc development[20]. **a** UMAP embeddings[50] following different data correction procedures (rows) and colored by different covariates, representing unwanted sources of variation and desired biological signal (columns). Rows (top to bottom) show uncorrected data (Original data), SiFT filtering using knn kernel (SiFT), Scanpy's scanpy.pp.regress_out() (regress_out), and scVI latent space with continuous covariates (scVI). Columns (left to right) show the cells colored by batch label, sex label, cell cycle phase, and *Nrt*, a novel Hedgehog target gene identified in ref. 20. **b** Integration and biological preservation scores per method. Scores (left to right): graph iLISI score evaluated for cell cycle phase label, graph iLISI score evaluated for cell cycle phase and sex label and overlap of the highly variable genes with a set of genes of biological interest. Scores are reported over *n* = 4 biologically independent samples. In box plot middle line, median; box boundary, interquartile range (IQR); whiskers, 1.5*IQR; minimum and maximum, not indicated in the box plot; gray dots, points beyond the minimum or maximum whisker ("Methods"). **c** The runtime of the methods on subsampled versions of the Heart Cell Atlas dataset[23]. **d** A table summarizing several criteria regarding the different methods: (top to bottom) adaptive; relates to the flexibility of the method, support of different filtering procedures allowing for optimization of the task. Both compared methods, regress_out and scVI, do not support additional parameters apart from the variable of interest, regarding the filtering task. runtime; color indicates the overall scalability of the method. A combined measurement of overall runtime and scalability across magnitudes, as depicted in (**c**). GPU support; implemented support of GPU acceleration. corrected expression; indicates whether the method outputs the corrected gene expression (for a disclaimer regarding scVI's applicability to this evaluation as it requires imputation of the corrected gene expression see "Methods"). Source data of (**b**, **c**) are provided as a Source Data file.

implementation of the linear regression suggested by Satija et al.[18] (regress_out[20]) and scVI conditioned on the expression of the set of nuisance genes (scVI[10]). This can be seen by qualitatively comparing the labels' separation in the latent space representation (Fig. 2a), and quantitatively with respect to the graph iLISI score (Fig. 2b).

Importantly, while filtering unwanted variance in the data SiFT preserves biological variation that was not targeted for filtering, with 40% of the genes of interest (using genes reported in the original analysis of this data[19] and in ref. 21, Supplementary Table 1) present in the corrected highly variable gene set, whereas regress_out captures

only 30% ("Methods", Fig. 2b). Such preservation of biological variation is visualized for Neurotactin (*Nrt*) and midline (*mid*), downstream Hedgehog pathway targets in the adult muscle precursors, and patched (*ptc*), a receptor of the Hedgehog ligand (Fig. 2a and Supplementary Fig. 2).

Further exploring this setting, when coupled to standard batch integration methods (e.g., Harmony[8] or bbknn[7]), over the complete integration task, SiFT positively compares to regress_out followed by Harmony or bbknn, or scVI applied with batch correction with continuous covariates (Supplementary Fig. 4). Next, we probe the dependence of SiFT on prior knowledge (the set of marker genes in this case). We evaluate the methods' performance using a partial, weaker set of markers, by randomly replacing 10 of the 55 markers with genes that are unrelated to the cell cycle and sex signal we aim to filter, yet related to biological signals we aim to retain post filtering (from the set of genes of interest). This exchange implies that beyond removing genes that are valuable for the recovery of the nuisance signal we add genes that encode biologically meaningful information, such as the spatial compartment in the wing disc, thus weakening the gene set that is used to guide the filtering ("Methods"). In contrast to other methods, SiFT encounters a minimal degradation in performance (percentage of deviation of the mean score, taken over biological batches and modified marker set repetitions, from the score over the original set: iLISI cell cycle; SiFT = −0.6%, regress_out = −1.0%, scVI = −13.5%, iLISI cell cycle and sex; SiFT = −4.2%, regress_out = −5.9%, scVI = −16.3%, Supplementary Fig. 4).

At last, we turn to show that SiFT is scalable and can be applied to large single-cell datasets. Of note, similarly to scVI, SiFT's scalability relies on GPU support (see "Methods"). Runtime performance is benchmarked on the Heart Cell Atlas dataset, composed of nearly 500,000 cells[22]. To define a filtering task, we add random features in the form of continuous random noise. The features are used as the mapping for the signal we wish to remove ("Methods"). Given this task we test the runtime of the removal algorithms (SiFT, scVI, and regress_out) over increasing subsamples of the data (Fig. 2c; both SiFT and scVI are run using GPU support). This showcases SiFT's efficiency and scalability to large single-cell datasets, with a runtime of 12 min, 36 min, and 1 h 26 min for SiFT, scVI, and regress_out, respectively, over the complete dataset.

Together, SiFT meets all criteria desired for the successful filtering of unwanted signals in single-cell data (Fig. 2d); its versatility in kernel computation allows adapting the filtering to uncover and preserve the biological signal of interest, its runtime efficiency using GPU support allows scaling to large datasets, and since filtering is performed directly on the input count matrix, it can naturally be incorporated into the data preprocessing step and followed by any downstream analysis procedure.

## SiFT exposes underlying biological heterogeneity

An important aspect of the filtering procedure is the ability to expose underlying biological attributes. The cell cycle introduces heterogeneity that can obscure other biological differences between cells, and so removing its effect improves the inference of inherent biological diversity[11–14,15]. Hence, many dedicated methods were introduced for this task, amongst them are Cyclum[11], an autoencoder-based approach for identifying circular trajectories in gene expression space, Seurat[13], which uses a linear model to find the relationship between gene expression levels and marker genes scores it assigns to each cell, ccRemover[14], a PCA-based method that identifies and then removes components related to the cell cycle, and f-scLVM[15], a factor-analysis-based latent variable model.

We consider a synthetic manipulation of single-cell data curated by Liang et al.[11] which consists of two sub-clones, and show that SiFT can successfully remove the cell cycle effect and enhance the subclone separability in the data. The two sub-clones provide a supervised setting resembling a biological signal we intend to preserve and expose. We used *mouse* embryonic stem cell (mESC) data as one clone[23] and a second clone was created by doubling the expression levels of a randomly selected set of genes containing variable numbers of cell cycle and non-cell cycle-related genes. The cell cycle stage can be easily recovered in the original data, but not in the subclone where it is hidden (Fig. 3a).

To quantify the performance and assess the separability of the sub-clones along with the mixing of the cell cycle stages, we use a set of metrics for integration accuracy[24]. The metrics are divided into two categories, removal of batch effects and conservation of biological variance (all scores are scaled between 0 to 1), where cell cycle stage labels are considered as the batches in the data. Hence in our context, the batch effect removal scores correspond to the mixing of the cell cycle stages. Similarly, the cell subclone identity label is used as the anchor for the preservation of biological signals, thus used in the metrics for biological variance ("Methods").

We start with an independent evaluation of SiFT using only a set of known cell cycle marker genes as input to compute a knn kernel ("Methods", Fig. 3b). The marker genes define the manifold for the computation of the neighbors. Here, the low-dimensional representation of the cells does not expose clear visual separation between the sub-clones (Fig. 3c), yet the quantitative assessment ensures it performs well for the desired task and more specifically outperforms Seurat which relies on the same prior knowledge (Fig. 3d). Next, we show how additional prior knowledge can enhance the accuracy of SiFT by two alternative mappings for the cell cycle signal. The ground truth cell cycle stage (provided with the data), and the binned representation of the pseudotime inferred by Cyclum[11]. Both are then used to construct mapping kernels. The mapping kernels showcase a block structure depicting three cell cycle stages while the knn kernel captures more subtle relations between the cells (Fig. 3b).

Further, we evaluate the performance of SiFT using three additional kernels: A distance kernel and a coarser knn kernel on Cyclum's pseudotime, as well as a mapping kernel based on Seurat's[13] inferred cell cycle labels. SiFT preserves its performance accuracy using the distance kernel over the Cyclum pseudotime, whereas the coarser binning of the pseudotime slightly degrades the results, still comparing favorably to all methods apart from Cyclum. However, as a direct result of SiFT's dependence on the accuracy of the input mapping, using Seurat-based labels results in a loss of accuracy (Supplementary Fig. 5).

Following filtering of cell cycle effects, the visual separability of the sub-clones is substantially enhanced by SiFT, as well as by Cyclum, using either the ground truth labels or Cyclum's pseudotime (Fig. 3c). SiFT attains quantitatively higher scores in cell cycle removal metrics in comparison to baseline methods, and higher scores in biological conservation when relying either on ground truth or Cyclum based mapping to be filtered (Fig. 3d and Supplementary Fig. 5).

## Filtering spatial signals recovers temporal information

Cellular gene expression is regulated by spatial and temporal signals, posing a challenge of decoupling, and studying the interplay between the two. The liver stands as an example of a tissue undergoing strong spatial and temporal regulation; it consists of repeating anatomical units termed liver lobules, and sub-lobule zones performing distinct functions. Liver zonation refers to functions that are non-uniformly distributed along the lobule radial axis. Hence, even for spatially resolved data, recovering the spatial zonation requires identifying the lobules and classifying cells into their sub-lobular zones. Given the strong zonation signal, regardless of spatial information this is often done using known marker genes. Beyond the spatial structure, the liver is also subject to temporal regulation, consisting of the circadian clock, systemic signals, and feeding rhythms[25,26]. While the liver zonation signal has been intensively studied[26,27], less is known regarding the

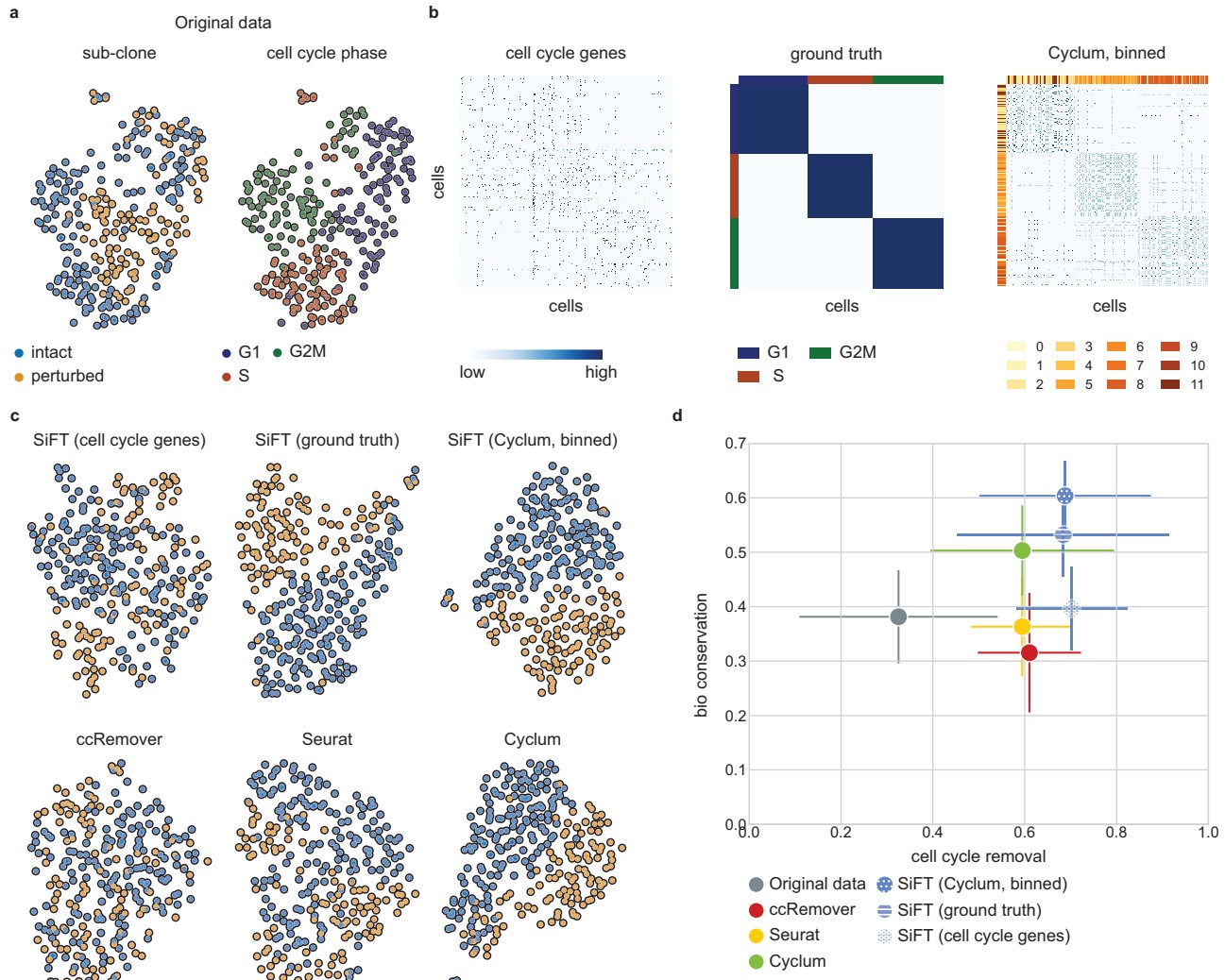

**Fig. 3 | Filtering the cell cycle effects from virtual tumor data consisting of two sub-clones. a** A UMAP of the original data colored by subclone identity (left) and cell cycle stage labels (right)[11]. **b** The different cell–cell similarity kernels as defined by SiFT, cells are ordered according to the ground truth cell cycle stage. (left) a knn kernel, neighbors defined using the gene expression of a set of cell cycle marker genes. (center) a mapping kernel based on the ground truth cell cycle stage, the row (col) colors depict the cells' label. (right) a mapping kernel based on binning of the Cyclum pseudotime, the row (col) colors depicts the cells' bin. **c** UMAP of the filtered data colored by the subclone identity. The top row presents SiFT filtered data, (left) cell cycle marker genes (center) ground truth labels (right) Cyclum pseudotime binning Bottom row presents compared methods, (left) ccRemover (center) Seurat (right) Cyclum. **d** Scatter plot of the mean overall bio conservation score against mean overall cell cycle removal scores using the metrics defined in ref. 25 ("Methods"). Error bars indicate the mean standard error considering $n = 6$ bio conservation metrics ($y$ axis) and $n = 4$ cell cycle removal metrics ($x$ axis). Source data of (**d**) are provided as a Source Data file.

temporal signal[25]. Thus, SiFT can be used to filter the prior knowledge regarding the spatial zonation signal in order to enhance the circadian clock trajectory. To do so, we consider a dataset from ref. 26, labeled for temporal processes, as data were collected at four different equally spaced time points along the day, and with known marker genes for the spatial zonation signal[27] (Supplementary Table 2). The prior knowledge of known marker genes allows to computationally recover the spatial structure for filtering (Fig. 4a).

While SiFT can take as input the set of spatial marker genes directly, it can be beneficial to use existing methods dedicated to spatial reconstruction and provide the spatial mapping directly as input to SiFT. Using SiFT solely with marker genes exposes the temporal signal and by providing the spatial reconstruction input, we can further improve its performance. To obtain a spatial reconstruction we use novoSpaRc[2,3], providing a mapping of the spatial organization of the cells. NovoSpaRc is an optimal transport-based method for reconstruction of scRNA-seq data which can take as input prior knowledge in the form of marker genes ("Methods"). The mapping, as

inferred by novoSpaRc, to an eight-layer tissue representing the lobular liver layers, is used by SiFT to construct a mapping kernel (Supplementary Fig. 6). The set of spatial marker genes is used by SiFT to define a knn kernel, where distances between cells are computed based on similarities in the spatially zonated genes (Supplementary Fig. 6).

In both settings, using zonated genes or relying on novoSpaRc mapping, applying SiFT successfully removes the zonation signature yet preserved the visual separability based on the circadian trajectory over a UMAP representation of the data (Fig. 4b). This is further supported by comparing the expression of two zonated and rhythmic genes (*Hlf* and *Elovl3*) before and after the application of SiFT (Fig. 4c); The genes' temporal trend is preserved (Fig. 4c, top row), yet the spatial variation is eliminated (Fig. 4c, bottom row).

We quantitatively test the performance of temporal reconstruction by assessing the correlation of the reconstruction with the reported temporal trajectory as a function of the number of temporally informative genes (marker genes used for the

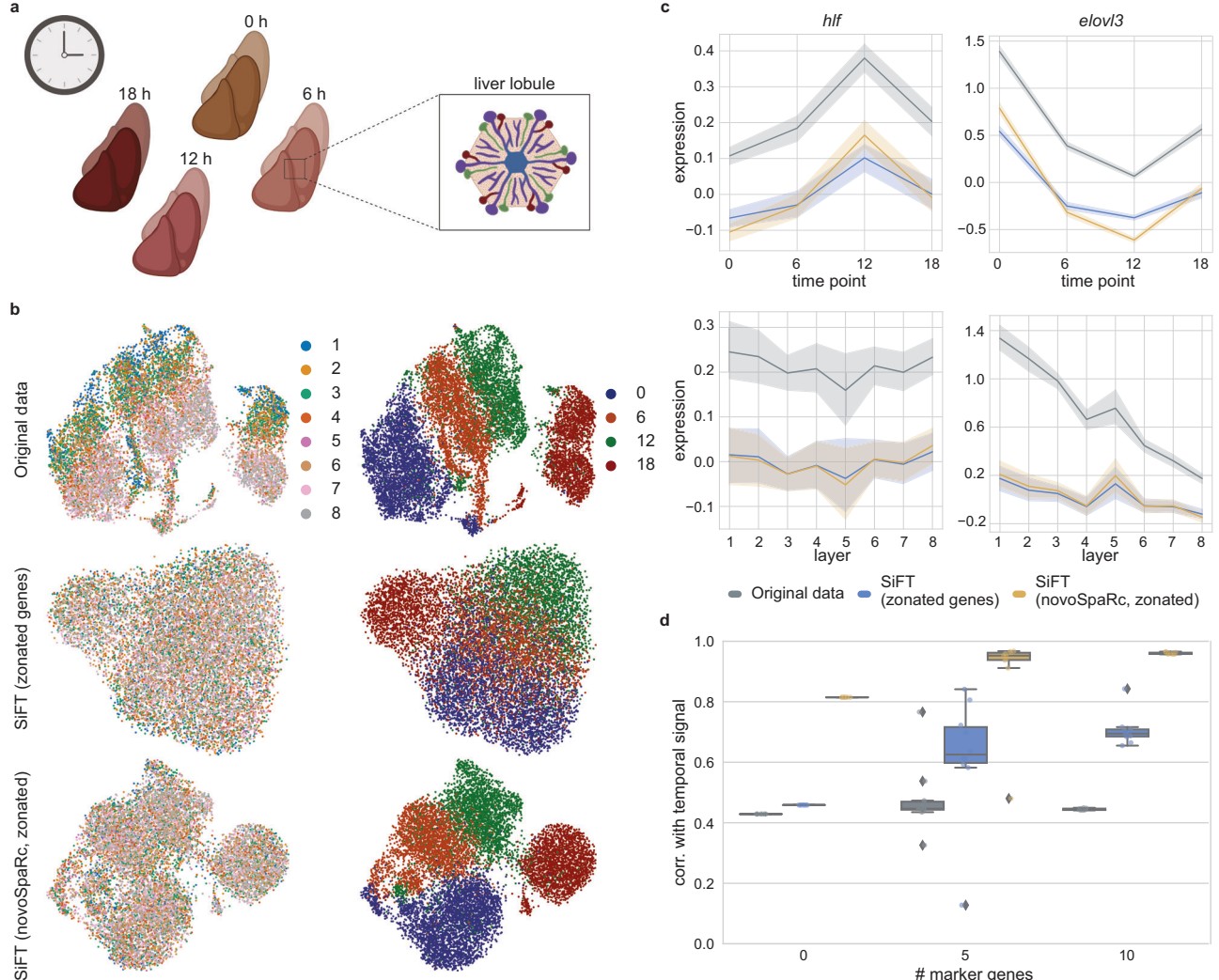

**Fig. 4 | Enhancing circadian clock signal in the mammalian liver by filtering spatial zonation signals. a** Single-cell RNA-seq of hepatocytes isolated at four different times within a day[26]. Figure is created with BioRender.com. **b** A UMAP representation of the original data (top), SiFT filtered data based on zonated genes (middle), and novoSpaRc zonated mapping (bottom). Plots are colored by the lobular layer (left) and time point (right). **c** Reconstructed temporal (top) and spatial (bottom) expression patterns of zonated and rhythmic genes (*hlf* and *elovl3*). Solid line depicts mean expression and bandwidth represents the 95%

confidence interval over $n = 14364$ cells. **d** Pearson correlation between reconstructed and original rhythmic signals as a function of the number of temporal reference genes used for the reconstruction (Pearson correlation is evaluated over $n = 20$ rhythmic informative genes, results present the mean value over $n = 10$ independent repetitions of novoSpaRc). In the box plot middle line, median; box boundary, interquartile range (IQR); whiskers, 1.5*IQR; minimum and maximum, not indicated in the box plot; gray dots, points beyond the minimum or maximum whisker. Source data of (**c**, **d**) are provided as a Source Data file.

reconstruction, Supplementary Table 2). To do so, we employ novoSpaRc again, this time to recover the temporal trajectory by mapping to four cyclic locations. We find that the quality of temporal trajectory reconstruction with SiFT (using novoSpaRc, zonated), even without any temporal reference, is better than the reconstruction based on the original data with 10 marker genes (Pearson correlation; SiFT (0 markers) = 0.81 ± 0.00, Original (10 markers) = 0.44 ± 0.00), implying that by using SiFT, prior knowledge of the zonation signature exposes the underlying circadian clock trajectory and provides sufficient information for its successful recovery. Further, even in the simpler setting, applying SiFT using zonated genes, improves the quality of temporal reconstruction without any marker genes (Pearson correlation=0.46 ± 0.0). At last, in both of the SiFT settings, the reconstruction quality substantially improves upon the addition of informative genes and reaches near-perfect performance using 10 markers (Pearson correlation; SiFT (zonated genes) = 0.71 ± 0.00, SiFT (novoSpaRc, zonated) = 0.96 ± 0.00, Original=0.44 ± 0.00, Fig. 4d).

## Recovering collective subpopulations by removing local interactions

A cell's state is largely influenced by its environment, and cell–cell interactions give rise to tissue niches[28]. While deciphering the communication mechanisms and emergent local spatial structure is crucial[28,29], understanding global cellular states is often the primary focus of interest. Advancements in methods for spatial molecular profiling provide us with contextual information regarding the cell's local environment along with gene expression measurements. As we show, the spatial information can be utilized to filter local signals and expose global phenomena.

As presented earlier, the liver consists of repeating anatomical units, inducing global subpopulations of cells acting similarly across a tissue sample. However, signals associated with such spatially global processes may be interfered with local spatial signals, making processes occurring at longer length scales harder to recover. Recently Matchett et al.[30] presented a cross-species, integrative multimodal dataset to study liver regeneration. Among their key reported findings

were the discovery of a novel ANXA2[+] migratory hepatocyte subpopulation emerging during human liver regeneration and a corollary subpopulation in APAP-induced *mouse* liver injury. For the analysis of the *mouse* model of APAP-induced acute liver injury, Matchett et al.[30] considered both snRNA-seq and spatial transcriptomics (ST) of *mouse* liver across multiple time points (Fig. 5a). Our analysis of the ST data shows that the observed expression patterns are largely driven by local neighborhoods, giving rise to spatially distinct localized clusters, while signals related to global subpopulations cannot be recovered. This is evaluated visually, by the spatial distribution of the clusters, and quantitatively through the clusters' interaction matrix over spatial coordinates (Fig. 5b, Supplementary Fig. 7, "Methods"). Specifically, the migratory subpopulation of cells cannot be recovered from ST

data directly without specific prior knowledge over the expected migratory signal (marker genes identified over human samples[30]). Since our initial analysis suggests that cellular gene expression is strongly driven by local information, we hypothesized that by removing this signal, which is encoded by spatial coordinates, we will be able to directly uncover the migratory hepatocyte subpopulation.

To this end, we apply SiFT to each ST slide independently, constructing a distance kernel based on the spatial coordinates ("Methods"). After filtering the local spatial signal, we perform leiden clustering of cells from all samples jointly. The obtained clusters expose global spatial patterns, shared across samples (Fig. 5b and Supplementary Fig. 8). Quantifying the distribution of samples within each cluster, we observe a relatively homogeneous distribution over

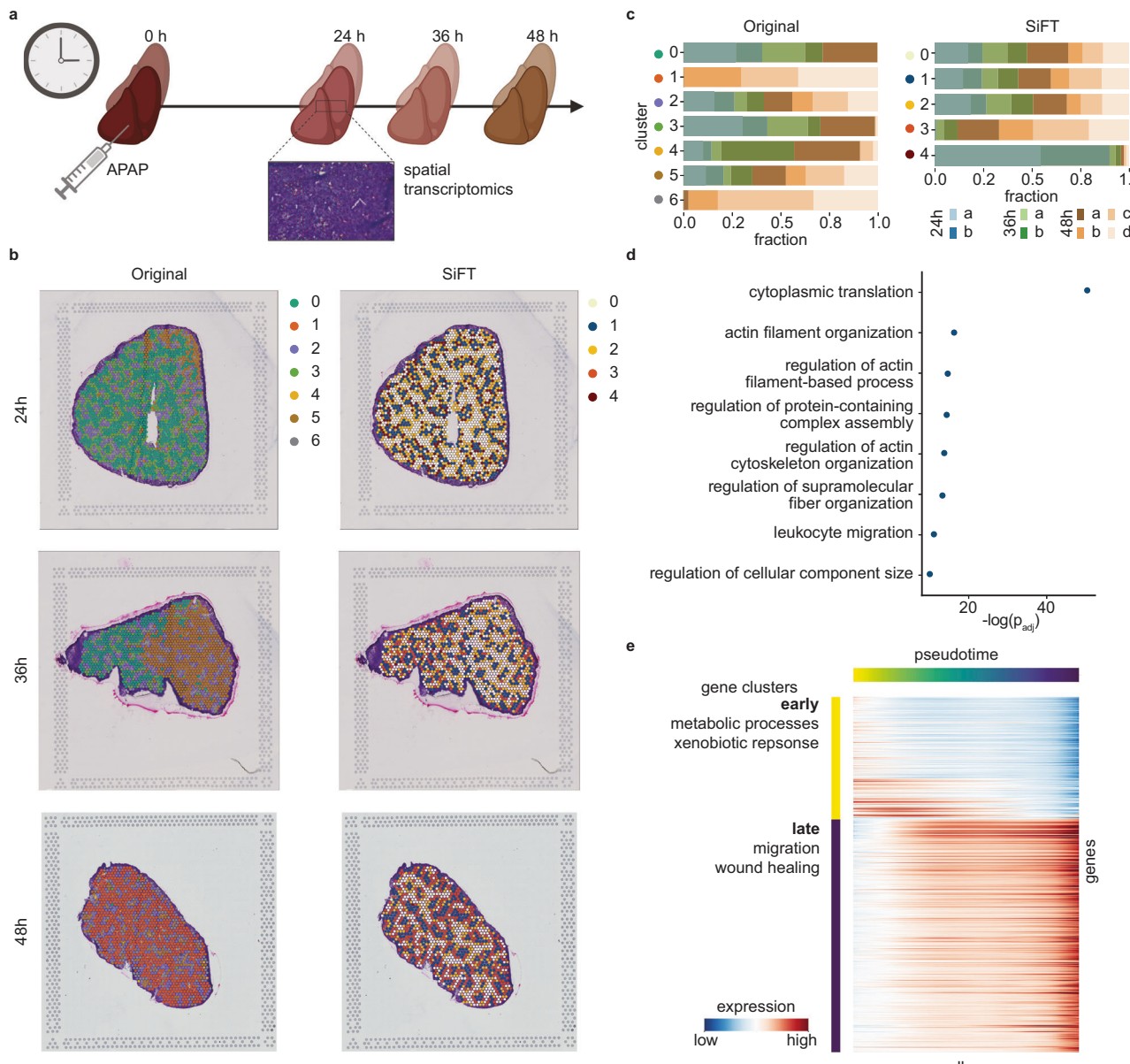

**Fig. 5 | Exposing collective subpopulations in spatial transcriptomics (ST) samples of liver regeneration. a** schematics of time points processed for ST measurements post APAP-induced liver injury in mice[31]. Figure is created with BioRender.com. **b** Spatial visualization of the leiden clusters identified over the original data (left column) and SiFTed data (right column). Rows are representative samples of the different time points, 24, 36 and 48 h (top to bottom). **c** The distribution of samples within each cluster with respect to clustering of the original

data (left) and SiFTed data (right). **d** GO terms enriched in the migrating cells subpopulation. **e** Pseudotime analysis of the migrating cells subpopulation recovers temporally correlated genes. The set of genes can be clustered into two groups, early and late, according to the population of cells along the pseudotime at which they are more highly expressed. Representative differential GO terms between early and late gene clusters are shown to the left of the heatmap. Source data of (**c**, **d**) are provided as a Source Data file.

the first three clusters and the two last clusters capturing distinct states in late and early samples, respectively. This contrasts with the non-homogeneous distribution of samples across the original clusters (without SiFT, Fig. 5c). Importantly, cluster 1 in the SiFTed data corresponds to the ANXA2+ migratory hepatocyte subpopulation, with *Anxa2* appearing uniquely in cluster 1's top marker genes (Supplementary Fig. 9) and corresponding GO terms suggesting a migratory phenotype (Fig. 5d). As mentioned above, this migratory subpopulation cannot be directly recovered by clustering of the original data[30].

After identifying the migratory hepatocyte subpopulation, we turn to study its temporal signature. Using scFates[31] we obtain a pseudotime ordering of the cells and identify temporally informative genes. Within the temporally correlated genes we can identify two subgroups, early and late, peaking at early (late) stages of the trajectory, accordingly (Fig. 5e). Using gene-set enrichment analysis, we find that genes peaking at late stages of the trajectory are related to wound closure, chemotaxis, and migration, in accordance with the identification of Matchett et al.[30], based, in that case, on additional experimental measurements. Downregulated programs at late stages consist of metabolic processes, which can be related to the expected degradation of metabolic processes in proliferating hepatocytes, which appear following wound healing[30,32] (Supplementary Fig. 10).

## Using a reference of healthy control cells to recover disease signature

An essential step in studying disease is decoupling disease signatures from the healthy state by characterization of the healthy signal in clinical samples. SiFT can be used for this task, recovering disease-specific signatures in individual cells, by filtering the healthy trajectory based on healthy control samples. This allows for enhancing the disease signature, identifying cells that are more informative for disease states, and studying the different types of disease response pathways (Fig. 1). Existing analysis pipelines approach this task by integrating all samples, from both healthy controls and disease patients, and then performing comparative analysis, for example, by performing differential expression analysis between common cell types or assessing differences in cellular composition between health and disease[33–36]. This analysis holds some limitations as it relies on global comparison between the groups and does not allow studying the disease trajectory in individual cells. In contrast to this, in the analysis performed using SiFT, the healthy trajectory in each disease cell is identified independently by weighting the contribution of control cells according to their similarity to it.

We considered single-cell transcriptomes from peripheral blood mononuclear cells (PBMCs) from individuals with asymptomatic, mild, moderate, severe, and critical COVID-19 ($n = 90$ individuals) and controls ($n = 23$ individuals) reported by Stephenson et al.[33] (Fig. 6a). We used harmonized PCA space (as reported in ref. 34), which corrects for batch effects between samples to define a knn kernel, capturing the similarity between each cell from individuals with COVID-19 relative to the healthy population (Fig. 6b. the robustness of the kernel choice was validated by evaluating performance also using the distance kernel, Supplementary Fig. 11, "Methods"). Under this construction, applying SiFT is expected to filter the signal of the healthy trajectory from the data and expose the disease response in each cell)

Patients with COVID-19 present with an abnormal immune landscape, characterized by overactivated inflammatory, innate immune response, and impaired protective, adaptive immune response[37]. Recent studies revealed the dynamic changes in peripheral immune cells, both in transcriptional states and population size over the course of COVID-19[34,38]. The SiFTed representation recovered cell types involved in innate immune response, based on the interferon response (IFN) score (monocytes, DCs, and HSPC), whereas the same analysis over the original data failed to expose the relevant cell types[33] (Fig. 6c). While this acts as validation that SiFT can successfully recover results

obtained from direct comparative analysis[33], we next show how using SiFT can go beyond current analysis methods.

Considerable effort has been put into identifying the expansion of different cell types in response to COVID-19 infection[33,39]. This is assessed by comparing the cell-type-specific population size between the disease and control samples. This analysis, however, does not expose the extent to which these cells respond to the infection and furthermore, which fraction of the expanded population contains information regarding the disease state. SiFT allows for both as it refines and extends the initial cell-type classification with respect to the disease response, hence exposing the distinct underlying disease signature and identifying the cells that are informative for the disease state. Intuitively, by SiFTing the healthy signal we expect that cells of a given cell type with a dominant disease response will preserve their identity (and will later be clustered together) and that remaining cells, with a less distinct response (more similar to healthy cells), will tend to cluster with non-informative cells, as most of their signal was removed. Indeed, the latent space representation of the SiFTed data exposes subpopulations of specific cell types (CD14+ mono., CD16+ mono., Plasmablast, and Platelet) while other cell types got mixed (Fig. 6d). Importantly, the exposed cell types are known to undergo changes in response to COVID-19 infection[40].

Next, to obtain a classification of cells as disease-informative, we used a cluster-purity test over Leiden-based clusters in the SiFTed data with respect to cell-type labels ("Methods", Supplementary Fig. 12, Fig. 7a). Under the cluster-purity test, a cluster is classified as disease-informative if the fraction of the most prominent cell type in it exceeds a threshold value $\tau = 0.55$. Enrichment analysis over the differentially expressed genes between the two groups (disease-informative/disease non-informative) supported this classification and identified pathways associated with inflammation including response to virus, response to type I interferon, IFN$\alpha$ response, inflammatory response, and regulation of viral process in the disease-informative cells (Fig. 7b). In addition, under the classification of disease-informative cells, the IFN response signature was refined by enhancing the expected pattern of disease informative cells and exemplifying the lack of disease-related signal of remaining disease non-informative cells (Supplementary Fig. 12). In accordance with this, the disease-informative cells exhibited overexpression of type I/III interferon response-related genes (Supplementary Fig. 12), which were recently reported in genome-wide association studies (GWAS) for COVID-19 susceptibility[41,42].

The classification of disease-informative/non-informative clusters follows the understanding that analysis based on cell-type abundance is limited and may not accurately reflect the relevance for the underlying disease[43,44]. For example, there is an ongoing debate regarding the influence of SARS-CoV-2 on the HSPC niche[43] and it has been shown that HSPCs are particularly susceptible, implying that their vulnerability to SARS-CoV-2 may vary greatly from patient to patient[45]. Thus, despite being the fourth most prevalent cell type in clinical samples it may be that only a small subset of these cells is informative for disease state, as indicated by the SiFT clustering (Fig. 7c). On the other hand, within the cDC population, ranked third least prevalent cell type in clinical samples, most of the cells were found to be associated with the disease-informative cluster. This supports the importance of cDCs in COVID-19 response, as expected since cells of myeloid origin play a pivotal role during infections. Specifically, there is growing evidence that even though their fraction in disease state decreases, cDCs undergo aberrant maturation in COVID-19[44,46,47], amongst them are recent findings by Marongiu et al.[44] that SARS-CoV-2 interacts directly with this population. We can study the disease-associated signal in this population by performing differential gene expression analysis between the disease-informative and non-informative clusters. This recovers the importance of this population in virus response, and highlights previously reported genes, *IFITM* genes and *MX1*[44] (Fig. 7a and Supplementary Fig. 13).

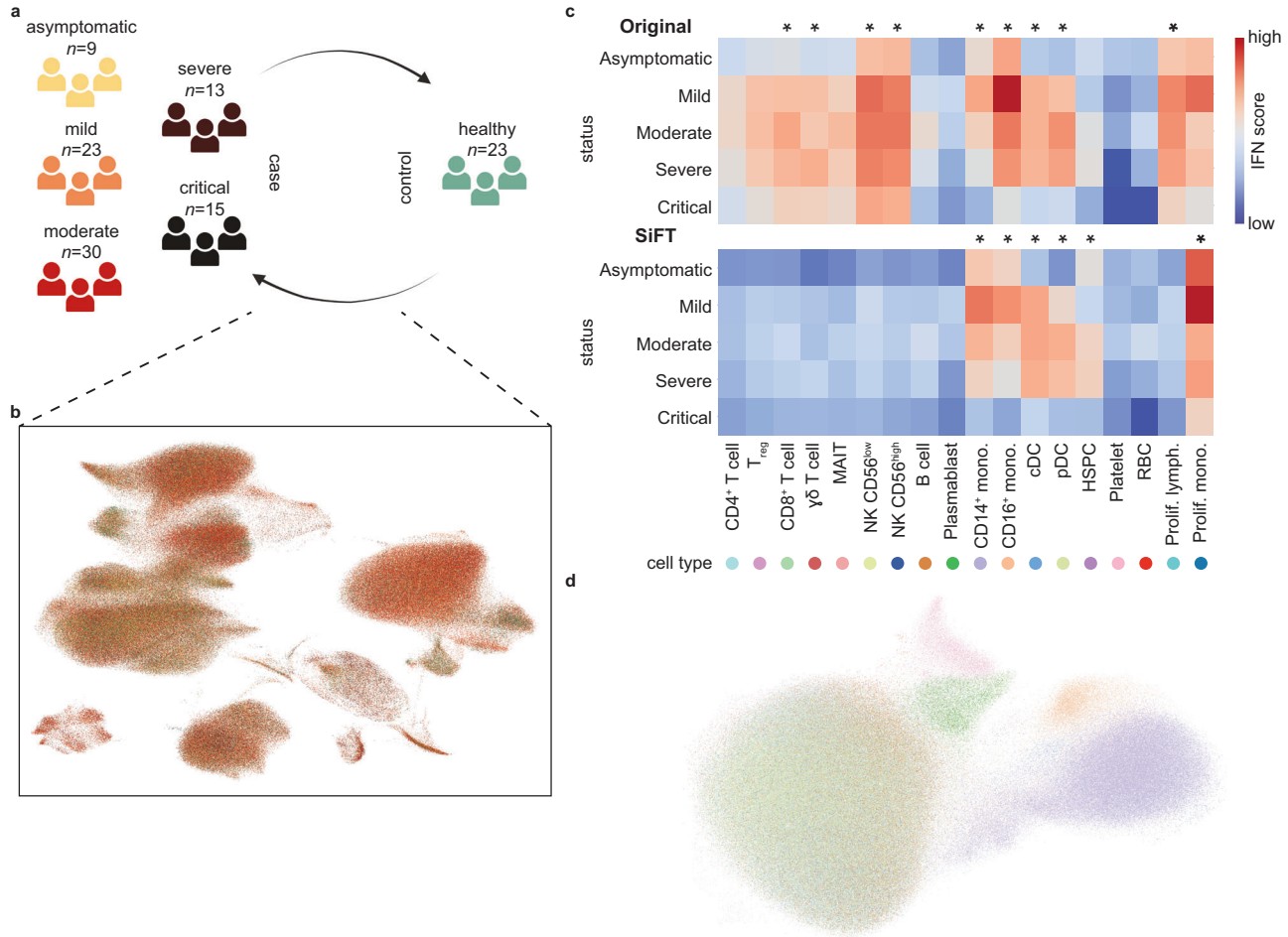

**Fig. 6 | Exposing disease signature through the removal of similarity to healthy control cells in single-cell PBMCs transcriptomes from individuals with asymptomatic, mild, moderate, severe, and critical COVID-19. a** Overview of the participants included, and the data collected by Stephenson et al.[34]. A total of $n = 90$ COVID-19 patients and $n = 23$ control individuals. Figure is created with BioRender.com. **b** A UMAP representation of the original data colored by disease state. The cells from healthy samples are used as reference for SiFT filtering. **c** Enrichment of interferon response of each cell state separated by disease severity. Shown for original data (top) and SiFTed data (bottom). IFN response score was calculated using a published gene list (GO:0034340). Statistical tests were performed with a one-sided Mann–Whitney $U$ test between the cell types. Cell types are considered statistically significant if $p_{val} < 0.05$ (denoted by *). **d** UMAP visualizations of disease cells in the SiFTed data colored by reported cell type. Source data of (**c**) are provided as a Source Data file.

In addition to exposing cell types of interest, this analysis identifies the population of interest within a cell type. For example, while increased platelet activation and coagulation abnormalities were previously reported in COVID-19 patients[33,48], we identified two distinct subpopulations of Platelet: informative and non-informative for disease state (Fig. 7a). Enrichment analysis based on differential gene expression between the disease-informative and non-informative clusters identified pathways associated with coagulation, hemostasis, and antimicrobial humoral response in the disease-informative cells, as well as increased expression of surface proteins indicative of platelet activation in the disease-informative cells (Supplementary Fig. 13).

## Discussion

We presented SiFT, a method aiming at discovering hidden cellular processes by filtering out a known or reconstructed signal from single-cell gene expression data. The SiFT procedure starts by defining a cell–cell similarity kernel, capturing similarities with respect to the signal to be filtered. This kernel is then used to obtain a projection of gene expression onto the signal, which is then removed from the original expression.

We have shown that filtering signals by SiFT can expose the underlying, biologically meaningful structure in the data over a wide range of tasks. First, we showcased its ability to successfully filter unwanted sources of variation caused by nuisance signals in the data while preserving biological signals of interest. When focusing on removing cell cycle effects, a major source of bias in single-cell data, in a semi-simulated setting, we showed that SiFT outperforms state-of-the-art methods for the task. A substantial advantage of SiFT is its ability to use existing prior knowledge to reveal hidden biological attributes. We used the vast prior knowledge regarding spatial zonation in hepatocytes to uncover the temporal trajectory in the data. In addition, we applied SiFT to spatially resolved dataset of liver regeneration, where we used the spatial coordinates to remove local signals and expose global subpopulations of interest involved in the regeneration process. At last, we presented SiFT's applicability to the case–control setting. In the context of COVID-19, SiFT exposed disease-related signals and single-cell dynamics by filtering a corresponding healthy trajectory obtained by mapping to reference healthy samples.

In contrast to various latent space representation methods, SiFT performs the correction at the level of individual genes. In turn, similarly to other data correction methods, it modifies the gene expression so that data is no longer properly log-transformed, a property that is expected in certain downstream analysis tasks. However, this can be

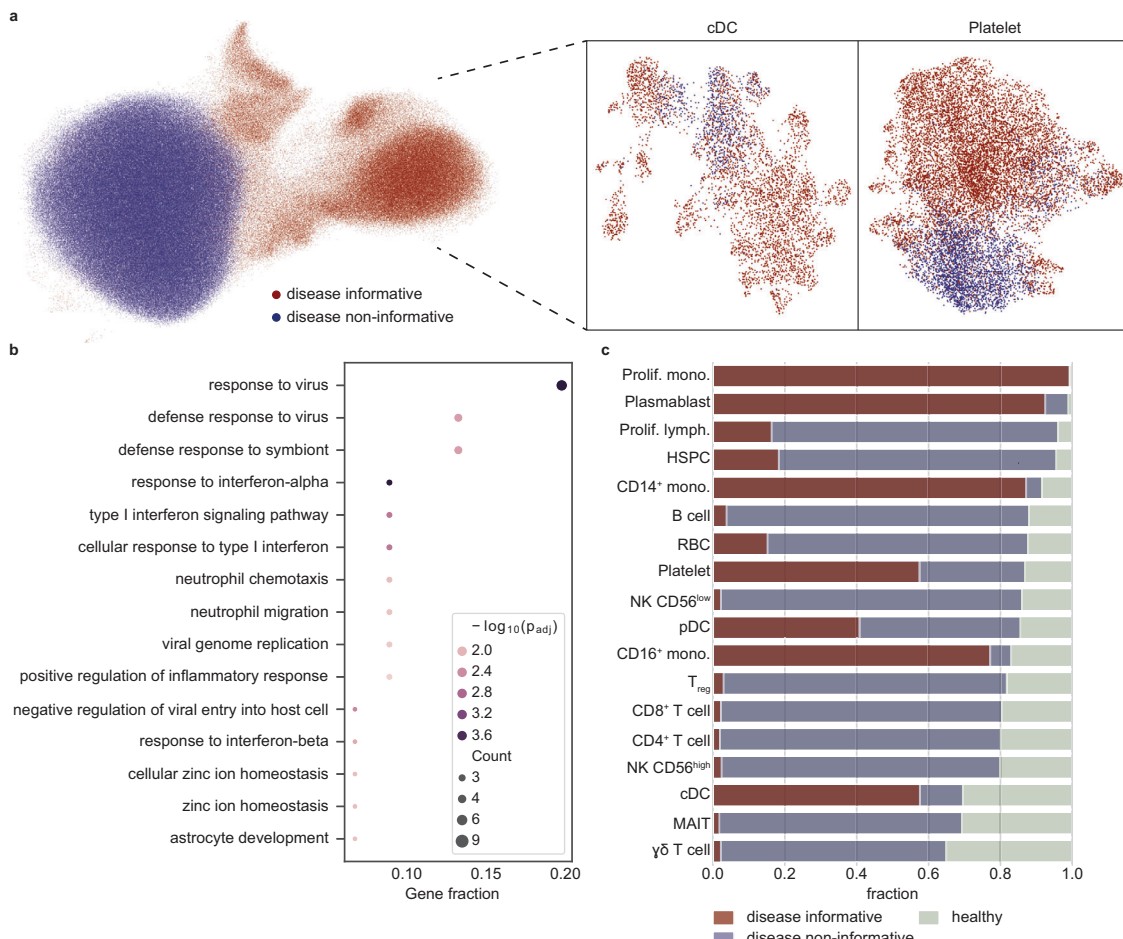

**Fig. 7 | Classification of cells from COVID-19 clinical samples to disease-informative (or non-informative). a** UMAP visualizations of all cells in the SiFTed data colored by association to disease-informative and non-informative clusters. Leiden clusters in the SiFTed data were classified according to cluster-purity score ("Methods"). Insets show a zoom-in on the cDC (left) and Platelet (right) sub-populations. **b** Enrichment analysis of the differentially expressed genes in the disease-informative cluster (compared to non-informative, using top 50 genes).

The size of the circles indicates the number of genes. Color indicates the magnitude of $-\log_{10}\left(p_{adj}\right)$. $p_{adj}$ is calculated using the hypergeometric distribution, a one-sided version of Fisher's exact test, with Benjamini–Hochberg correction. **c** Disease-informative bar plot of the proportion of cell populations, separated into disease-informative, disease non-informative, and healthy (according to the assignment shown in (**a**)). Cell types are sorted according to the fraction of disease cells. Source data of (**b**, **c**) are provided as a Source Data file.

corrected by adding a global pseudocount which preserves the relative differences in gene expression. Performing the correction directly on the measurements is a desired property for many biological applications as it allows drawing interpretable insight at the gene level, rather than a set of lower dimensional inferred features. In future work, it will be intriguing to explore the statistical implications of applying commonly used analysis tools to corrected gene expression data.

SiFT suffers from the general challenge of any computational method which lacks a ground truth reference that can be used to evaluate the results. To mitigate this, we suggest using integration metrics (which in general may be application-specific), as has been used throughout this manuscript, which provides a quantitative evaluation of the data after filtering is applied. Finally, it is important to note that SiFT strongly relies on the prior knowledge used to define the cell–cell similarity kernel. As we showcase in various examples, an inaccurate or coarse-grained mapping may lead to a loss of signal.

Here we exemplified the diverse range of applications of SiFT and showcased the potential that filtering has for understanding the different biological signals encoded in single-cell data. We further envision that with the ongoing increase in single-cell analysis tools along with the advance in multimodal assays, SiFT will serve as a basic analysis tool revealing hidden, more complex structures in the data. We have made SiFT available as an open-source python package along with documentation and tutorials and ensured it can efficiently scale to the ever-increasing sizes of single-cell datasets.

## Methods
### The SiFT algorithm
The aim of SiFT is to expose hidden biological signals in an input count matrix. Given an expression matrix along with a mapping of the genes to a specific signal, SiFT computes the cell–cell similarity kernel based on this mapping, the projection of the data onto the signal, and the filtered expression matrix.

The input to SiFT includes a cell ($N$) by features ($G$) matrix $X \in \mathbb{R}^{N \times G}$ and a mapping of the cells, $T$. For best performance, we recommend that the matrix $X$ will contain pre-processed normalized gene expression. In addition, SiFT can also be applied to any other representation of the cells, e.g., latent space or a subset of the genes.

The mapping, $T$, is assumed to capture a specific biological attribute or representation of the cells and can be of any type, e.g., stochastic or deterministic, continuous or discrete, uni- or multivariate. The diversity of types of mappings that are supported includes, for example, donor age (deterministic, discrete, univariate), pseudotime reconstruction (deterministic, continuous, univariate), and a latent space representation of the data (PCA, scVI, etc.)

(deterministic, discrete, multivariate). Alternatively, a set of process-specific marker genes (e.g., cell cycle marker genes), can be considered as a type of mapping and used to define distances between the cells. The type of mapping provided dictates the cell–cell similarity kernel that can be used.

The SiFT procedure comprises three main steps:

1. Compute cell–cell similarity kernel ($K$): In the first step, SiFT computes a cell–cell similarity kernel, $K \in [0,1]^{N \times N}$. The kernel is a row stochastic matrix, e.g., rows sum to one, hence defining a proper distribution. The details of the kernel construction depend on the specific kernel choice (see below). In all cases the user can choose the subset of source and target cells, that is the cells for which the kernel is computed (source) and the cells the similarity is evaluated with respect to (target). The user can also provide a pre-computed kernel.

2. Project the data ($P$): obtain a projection of the data on the supervised signal, $P \in \mathbb{R}^{N \times G}$

$$P = K \cdot X. \tag{1}$$

The stochasticity of $K$ guarantees that $P$ is of the same order of magnitude as $X$.

3. Filter the data (SiFTer, $\widetilde{X}$): deduct the projection, $P$, from the original count matrix, $X$, to obtain a filtered expression, $\widetilde{X} \in \mathbb{R}^{N \times G}$,

$$\widetilde{X} = X - P = X - K \cdot X. \tag{2}$$

## The kernels

The kernel, $K$, is supposed to capture the cell–cell similarity with respect to the signal we wish to SiFT. $K$ is a stochastic matrix such that the $i$ th row is a probability distribution denoting the similarity of cell $i$ to all observed cells. Broadly speaking, the kernels can be divided into three sub-classes, mapping, k-nearest-neighbor (knn), and distance kernel, differing in the type of prior knowledge or assumptions they require. Beyond the implemented kernels the user can provide a pre-computed kernel.

All kernels can be refined by restricting the source and/or target space. The source space relates to the cells whose expression we are interested in filtering. The target space is the cells over which similarity is assessed. This is done by specifying the sub-group of cells (e.g., only healthy cells in a disease-control experiment) of interest.

**Mapping kernel.** The basis of the mapping kernels is a stochastic or deterministic association of the cells ($N$) to a low-dimensional domain. The mapping, $T$ can be of any type, e.g., stochastic or deterministic, continuous or discrete, uni- or multivariate. In the case of a continuous variable a binning value $M$ is required, denoting the number of bins used to construct a binned representation. The mapping is represented by $T \in \mathbb{R}^{N \times M}$. Given $T$ we construct two sets of probability distributions:

1. $p_l(c_i)$: the probability of a label $\mu$ given cell $i$, normalizing $T$ across possible labels.
2. $p_c(\mu)$: the probability of observing cell $i$ given a label $\mu$, normalizing $T$ across cells.

The cell–cell similarity kernel, $K \in [0,1]^{N \times N}$, is defined as the multiplication between the two probabilities, $p_l$ and $p_c$, summing out the dimension of the embedded signal, $M$. Thus, each row in $K$ induces a normalized distribution, $p_{c_i}(c_j)$, defined for cell $i$ with respect to all cells in the dataset (or reference set),

$$K_{ij} = \sum_{\mu=1}^{M} p_l(c_i)p_c(\mu) = \mathbb{E}_{\mu \sim p_l}[p_c(\mu)] \equiv p_{c_i}(c_j). \tag{3}$$

As evident from the above construction, this kernel is most suitable when the prior mapping of cells is in a form of classification (of discrete classes or a continuum score) according to a known biological property. The quality of the filtering procedure will rely on the accuracy of the provided classification.

**K-nearest-neighbor (knn) kernel.** The mapping, $T \in R^{N \times M}$, is used to construct a neighborhood graph over the cells by finding the nearest neighbors for each cell. Thus, it is required that a distance metric, $D_{ij} = d(T_{i\cdot}, T_{j\cdot})$ (e.g., the Euclidean distance), is applicable to the provided mapping. We follow Scanpy's[20] defaults to set $k$ (with n_neighbors=15), and merge and process the neighbor sets via the UMAP algorithm[49]. Similarly to Scanpy[20] the user can adjust the choice of $k$, and as in other applications of the knn algorithm, modifying this parameter allows controlling the size of the neighborhood considered for the analysis, in this case the filtering procedure. In general, we show that the procedure is robust to this choice (Supplementary Fig. 3). At last, we normalize (across the rows) the resulting weighted adjacency matrix of the neighborhood graph of the cells (termed the connectivities) to obtain the final kernel object, $K$.

**Distance kernel.** Similarly to the knn kernel, a distance kernel requires that the provided mapping, $T \in \mathbb{R}^{N \times M}$, which defines a joint space representation of the cells, will be equipped with a distance metric, $D_{ij} = d(T_{i\cdot}, T_{j\cdot})$ (e.g., the Euclidean distance). With this, we define the cell–cell similarity kernel as,

$$K_{ij} = \frac{\exp\left(-\gamma d(T_{i\cdot}, T_{j\cdot})\right)}{Z_i}. \tag{4}$$

Here $\gamma$ is a smoothing parameter that controls the effective radii of cells (distances) that have a non-negligible influence, analogous to $k$ defined in the knn kernel. $Z$ is a normalization constant, defined for each row. If $d(\cdot, \cdot)$ is chosen to be the Euclidean distance, the above denotes the radial basis function (RBF) kernel, a popular kernel function used in various kernelized learning algorithms which can be interpreted as a similarity measure[50]. Similarly, if $d(\cdot, \cdot)$ is chosen to be the Manhattan distance, then $K$ is the Laplacian kernel.

**Determining kernel type.** As detailed above, kernel choice is largely guided by the form of the provided mapping, $T \in \mathbb{R}^{N \times M}$. Both the knn and distance kernel require that a distance metric could be defined over the mapping. Following this, as we demonstrate in the text (Supplementary Figs. 3 and 11), SiFT's performance is robust to the choice between these kernels, as well as to hyper-parameter choice. Therefore, while these choices can be avoided using the default setting of SiFT (the distance kernel with default parameters), we provide the option for both kernels and allow modifying their hyper-parameters for added user flexibility.

Importantly, the mapping kernel differs, as it is designed for a setting where the mapping provides an association of the cells to a low-dimensional domain which defines a notion of classification of the cells to different classes. Hence, while adequate for certain applications, for example filtering of cell cycle stage (Fig. 3 and Supplementary Fig. 5) it is not expected to perform well for a generic setting where the mapping does not capture a specific property. Further guidance and examples are provided in our online documentation, https://sift-sc.readthedocs.io.

## The stochastic interpretation of the SiFT algorithm and output

Looking into the mathematical derivation of the SiFT procedure we can expose natural stochastic properties which provide a better understanding of the SiFTed output. Recall that the input mapping, $T$, is used to define a row stochastic kernel, $K \in [0,1]^{N \times N}$. That is $\forall i \in$

$1, \cdots, N$ the $i$th row of $K$ is a probability distribution for cell $i$ with respect to remaining cells, we denote this distribution as $p_{c_i}(c_j)$

Now, an entry in the projected data $P_{ik}$ (for cell $i$ and gene $k$ in $P = K \cdot X$, step 2 in the SiFT procedure), can be read as,

$$P_{ik} = \sum_{j=1}^{N} p_{c_i}(c_j) X_{jk} = \mathbb{E}_{j \sim p_{c_i}} [X_{jk}]. \quad (5)$$

With this, for each cell, the projection $P \in \mathbb{R}^{N \times G}$, can be understood as the mean expression of genes according to the cell–cell similarity distribution. At the final step $P$ is deducted from the expression, so we obtain,

$$\widetilde{X} = X - \mathbb{E}_{p_c}[X]. \quad (6)$$

Namely, the filtered expression stands for the deviation of the cells' expression with respect to the expected value in their neighboring cells. This implies that the resulting matrix can contain negative values, which translate to genes (or features) that are below the average expression. We are only interested in the relations and distances between the cells in the new, filtered space, and not the absolute counts, hence the existence of negative values does not pose a problem. With that, realizing that certain analysis methods expect as input a positive count matrix, we suggest correcting the filtered expression matrix by adding a pseudocount following the global minima of the data (so that the corrected filtered minima would be zero), ensuring positivity and preserving the topology of the data.

## Pseudocount correction

We formulate here the implications of adding a pseudocount to the SiFTed expression to correct for negative values. Given $\widetilde{X} \in \mathbb{R}^{N \times G}$, the SiFTed expression, the pseudocount, $\alpha$ is given by, $\alpha = \min(\widetilde{X})$, and the SiFT pseudocount corrected expression, $\widetilde{X}^+$, $\widetilde{X}^+ = \widetilde{X} - \alpha$. This procedure ensures that $\forall x \in \widetilde{X}^+, x \geq 0$, and $\min(\widetilde{X}^+) = 0$.

Now, considering a single gene $i$, and SiFT pseudocount corrected expression $\widetilde{x}_i^+ \in \mathbb{R}^N$, the we have that the following relationships hold:

1. Mean ($\mu$): the mean of the SiFTed expression of the gene, $\widetilde{\mu}_i$, is shifted by a constant, the pseudocount value, $\alpha$, $\widetilde{\mu}_i^+ = \widetilde{\mu}_i - \alpha$.
2. Variance ($\sigma^2$): the variance is preserved,

$$\text{Var}\left[\widetilde{x}_i^+\right] = \mathbb{E}\left[\widetilde{x}_i^+ - \widetilde{\mu}_i^+\right] = \mathbb{E}\left[\widetilde{x}_i - \alpha - \widetilde{\mu}_i + \alpha\right] = \text{Var}\left[\widetilde{x}_i\right]. \quad (7)$$

3. Difference with gene $j$ ($\Delta_{ij}$): the difference between genes is preserved,

$$\Delta_{ij}^+ = \widetilde{x}_i^+ - \widetilde{x}_j^+ = \widetilde{x}_i - \alpha - \widetilde{x}_j + \alpha = \Delta_{ij}. \quad (8)$$

Relationships (1–3) imply that all metrics which rely on differences of values (or means) along with variance comparison are preserved under this correction. Among these are the standard $t$ test, and Wilcoxon signed-rank test.

## Runtime considerations

SiFT uses pyKeops, a python package allowing for Kernel Operations on the GPU without memory overflows[51] as a backend for matrix computations. The implementation supports pyKeops pytorch and numpy backends and hence does not enforce pytorch dependency. This implies that SiFT can scale to large datasets (Fig. 2c). Scaling relies on GPU support as the core of SiFT computation involves matrix multiplication over the cell's dimension, hence is quadratic in the dataset size. Since GPU memory is limited, to avoid falling back to CPU computation in the case of large datasets, SiFT performs automatic row-wise batching. That is, similarly to batches used in deep learning frameworks, SiFT automatically partitions the data to row-wise batches, and performs the computation using a GPU backend, retaining

speed performance. Importantly, this does not require any dedicated engagement from the user and is algorithmically valid since under the SiFT computation kernel rows are independent. In addition, whenever possible, computation is done over sparse matrices. For example, this may be the case if input count data is sparse and a knn kernel is defined over it. Again, this is the default performance of SiFT and does not require any additional input from the user.

## Datasets

**Drosophila wing disc myoblast cells.** We obtained the dataset of a temporal cell atlas of the *Drosophila* wing disc from two developmental time points collected in ref. 20 and using the processed data published in ref. 10, available at myoblasts.h5ad. The data contains two biological replicates were obtained at each time point that, after filtering for low-quality cells, generated data from 6922 and 7091 cells in the 96 hr samples and 7453 and 5550 cells in the 120 h samples.

To quantitatively assess our performance in removing unwanted variation with respect to these attributes, we turned to classify the cells into sex and cell cycle phase categories. The processed dataset was lacking sex and cell cycle labels however there are known marker genes for both. For sex labels, we follow the procedure suggested in the original study in which the data was presented[19]. The classification relies on the expression levels of the dosage compensation complex genes *lncRNA:roX1* and *lncRNA:roX2*. For both genes, we computed the density over the log-normalized counts and identified the first local minima as a threshold (Fig. 2 and Supplementary Fig. 1). Cells that were above the threshold for either *lncRNA:roX1* or *lncRNA:roX2* were classified as male; otherwise, they were classified as female.

Next, to obtain cell cycle phase categories we applied Scanpy's scanpy.tl.score_genes_cell_cycle() based on the expression of known *Drosophila* cell cycle genes from Tinyatlas at Github[52,53] (Fig. 2).

**Methods application.** To apply SiFT we defined a knn kernel using the set of cell cycle and sex marker genes (Supplementary Table 1). For the regress_out setting, Scanpy's scanpy.pp.regress_out() function was used by setting all marker genes as regression keys. For scVI we followed the reproducibility notebook, scvi_covariates.ipynb using only the set of cell cycle and sex marker genes as continuous covariates.

**scVI-corrected expression.** scVI provides the user with the option to impute normalized corrected counts through get_normalized_expression(). We use this function followed by sc.pp.log1p() to obtain the corrected counts used for HVG evaluation. It is important to note that scVI is not designed for this task precisely, hence, we evaluated its performance on the latent embedding when possible[10].

**Quantitative evaluation.** To evaluate the removal of cell cycle and sex signals we use the graph iLISI score as defined in ref. 25, using the corrected expression for SiFT and regress_out and over the latent embedding for scVI. We evaluated the graph iLISI score independently with respect to the "cell cycle phase label" and the combined "cell cycle phase and sex label".

To assess the perseverance of the biological signal of interest we use the set of genes of interest suggested in ref. 10 which is based on marker genes presented in refs. 20,22 (Supplementary Table 1). Here, we look at the intersection of this set with the set of highly variable genes (HVGs). HVGs identifies using Scanpy's scanpy.pp.highly_variable_genes() with n_top_genes=500 and flavor="cell_ranger". Since this cannot be done over the latent space we resort to using the imputed gene expression provided by scVI.

**Batch integration.** For additional evaluation regarding full integration performance following SiFT and regress_out, we applied batch integration methods. We used bbknn[7] and Harmony[8]. Both were run using Scanpy's functions with default parameters passing the batch key for

integration. For comparison, we use scVI with the batch label as a batch key along with the continuous covariates (scVI (covariates and batch)). Here, when computing the graph iLISI score, the latent embedding was used for all methods as neither bbknn nor Harmony provides a correction to the gene expression.

**Partial markers.** To assess robustness to the provided prior knowledge we test the performance of SiFT, regress_out and scVI using a partial set of marker genes. This set is partial in two ways: (1) 10 of the 55 original marker genes ("Sex genes" and "Cell cycle genes" in Supplementary Table 1) are removed and (2) 10 genes from the "Genes of interest" set (Supplementary Table 1) are included. By adding explicitly genes from the "Genes of interest" set we make the setting harder, ensuring that the set is not only partial in information with only 45 of 55 genes but that the additional 10 genes encode desirable biological signals which we wish to retain post filtering, such as the spatial compartment in the wing disc, and not just background noise. We perform the random sampling 10 times and provide all methods with the same modified set of genes.

**Heart Cell Atlas dataset.** The Heart Cell Atlas dataset was downloaded from https://www.heartcellatlas.org and contains a total of 486,134 cells.

**Benchmark sub-datasets.** For the runtime analysis, we followed the procedure suggested by Gayoso et al.[10]. Using the Heart Cell Atlas dataset which contains 486,134 cells, we created 8 datasets of increasing size by subsampling 5000, 10,000, 20,000, 40,000, 80,000, 160,000, 320,000, and 486,134 cells. For each dataset, the top 4000 genes were selected via scanpy.pp.highly_variable_genes(), with parameter flavor = "seurat_v3". Next, we generated 8 random covariates by sampling from a standard normal distribution and used them along with the percent_mito and percent_ribo fields as continuous covariates, defining a total of 10 continuous covariates.

**Methods runtime analysis.** Performed on NVIDIA RTX A5000 GPU. For SiFT runtime, we report the runtime of initialization of the distance kernel and running the filtering procedure. For scVI runtime, we report the runtime of the train function with the parameters used in ref. 10: early_stopping=True, early_stopping_patience=45, max_epochs=10,000, batch_size=1024, limit_train_batches=20, train_size=0.9 if n_cells <200,000 and train_size=1-(20,000/n_cells) otherwise. For the regress_out baseline, we tracked the runtime of the regress_out function for the above continuous covariates.

**Virtual tumor dataset.** The simulated dataset was downloaded from Cyclum's repository (https://github.com/Kchen-lab/Cyclum/tree/master/old-version/data/mESC). Details regarding the simulations of the virtual tumor data can be found in the original publication[11]. This data contains a total of 279 cells, 168 belong to the intact tumor and 111 to the perturbed. Cells are given with ground truth labels regarding the cell cycle phase.

**The kernels.** We consider different mappings representing the cell cycle signal. In total six different SiFT kernels are used for comparison:
- ground truth labels: A mapping kernel where the discrete cell labels are given by the ground truth cell cycle phase labels.
- cell cycle genes: A knn kernel where cell neighbors are computed based on the expression of the set of cell cycle genes reported in ref. 54.
- Cyclum pseudotime: we use Cyclum's pseudotime mapping of the cells to define three kernels:
  1. Cyclum, binned ($n = 12$): A mapping kernel where the discrete cell labels are obtained by binning the pseudotime to 12 bins.

  2. Cyclum, distance: A distance kernel where distances are computed over the mapping of the cells to the unit circle using the pseudotime, $(x,y) = (\cos(psd), \sin(psd))$.
  3. Cyclum, binned ($n = 3$): Same as above with three bins.
- Seurat: A mapping kernel where the discrete cell labels are given by Seurat's cell cycle phase prediction.

**Cell cycle removal methods.**
- Cyclum: we followed the steps in the provided example by the authors, example_mESC_simulated.ipynb. As the example is provided using an older version of Cyclum, we modified the parameters in the new implementation to correspond to the reported setup run in ref. 11.
- Seurat: we followed the steps suggested in the vignette cell_cycle_vignette.html.
- ccRemover: we used the ccRemover method as in the tutorial ccRemover_tutorial.html.
- f-scLVM: we used the slalom implementation of f-scLVM and followed the steps in the provided notebooks slalom/tree/master/ipynb.

**Evaluation metrics.** Evaluation metrics and their definitions were taken from ref. 25. We used the complementary python package scib (https://scib.readthedocs.io) which provides an implementation of all metrics. For the "cell cycle removal" score we reported the mean of ASW_label/batch, PCR_batch, cell_cycle_conservation, and iLISI_graph. For the "bio conservation" score we reported the mean of NMI_cluster/label, ARI_cluster/label, ASW_label, isolated_label_F1, isolated_label_silhouette, and cLISI_graph. All methods were run with default parameters.

**Mammalian liver dataset.** scRNA-seq data was downloaded from GEO, accession code GSE145197. For the data preprocessing procedure, we followed the pipeline described in the original publication[25] and given in the GitHub repository https://github.com/naef-lab/Circadian-zonation. After preprocessing the data contained 11,491 cells from 3 biological repetitions for 4 different time points (time point 0: 3563 cells, time point 6: 2791 cells, time point 12: 2919 cells, time point 18: 2218 cells).

**novoSpaRc for spatial mapping.** We used novoSpaRc[2,3] to recover the spatial signal in the data, and obtain a probabilistic mapping of cells to the eight liver zonation layers. We performed mapping using 15 spatial marker genes (reported in ref. 26) and ran the novoSpaRc algorithm with $\alpha = 0.5, \epsilon = 0.1$. We used the probabilistic mapping to apply SiFT (Supplementary Fig. 5).

**Liver regeneration dataset.** The liver regeneration multimodal dataset by Matchett et al.[30] is available under GEO, accession code GSE223561. For the mouse spatial transcriptomics data, we downloaded the provided Seurat objects (GSE223560_seurat_object.rds.gz) including 8 slides at three different time points relating to hours post injection (hpi). We first perform preprocessing to each sample independently using the counts layer. We follow basic preprocessing steps, filtering cells and genes, scanpy.filter_cells(min_genes=200), and scanpy.filter_genes(min_cells=3), normalize the data scanpy.normalize_total() and log transform, scanpy.log1p(). After preprocessing the dataset contained 14,798 cells distributed across samples as follows:
- 24 hpi: two samples of 1183 and 2258 cells.
- 36 hpi: two samples of 1444 and 1484 cells.
- 42 hpi: four samples of 1353, 2159, 2166, and 2747 cells.

**Distance kernel construction.** We apply SiFT to each sample independently, defining a distance kernel which is based on spatial distances between the cells.

**Leiden clustering.** The leiden scanpy implementation was used, scanpy.tl.leiden(resolution=0.5), the difference between the settings (Original and SiFTed) is in the cells adjacency used:

- Original data: To try to correct for sample effects, we use Harmony[30], correcting PCs over sample IDs. Then the adjacency used by leiden is the neighbors connectivities defined based on the Harmony PCs.
- SiFTed data: The adjacency used by leiden is the neighbors connectivities defined based PCs calculated over SiFTed data.

**Clusters' interaction matrix.** We use squidpy[55] to evaluate the spatial interaction of the leiden clusters within each sample. We first compute a connectivity matrix from spatial coordinates, squidpy.gr.spatial_neighbors(), and then use the dedicated function for interaction evaluation, squidpy.gr.interaction_matrix(key = "leiden").

**COVID-19 dataset.** The data object, as an h5ad file, was downloaded from haniffa21.processed.h5ad, haniffa21.processed.h5ad. Our analysis relied on provided data without further processing, to keep it as comparable as possible to original analysis of the data. In the pre-processing steps reported by Stephenson et al.[33], harmony was used to adjust PCs by sample ID. Given this, we used the provided Harmony PCs and no further batch correction or integration steps were performed. At last, we relied on cell-type labels and metadata reported in the given file. We considered COVID-19 samples, $n_{disease}$ = 527,286, and healthy controls, $n_{healthy}$ = 97,039.

**Knn kernel construction.** To filter the healthy trajectory, we define a knn kernel $n_{disease} \times n_{healthy}$ that captures the similarity of the disease cells to the reference healthy cells. As a basis to evaluate the similarity (distances) between the disease and healthy cells we consider distances in the harmonized PCA space, as elaborated above (reported in ref. 34). We use the default numbers of neighbors, n_neighbors=15.

**Distance kernel construction.** Similarly to the construction of the knn kernel we define a distance kernel $n_{disease} \times n_{healthy}$ that captures the similarity of the disease cells to the reference healthy cells using distances in the harmonized PCA space. To avoid numerical instabilities, as the distance kernel requires exponentiation of the distances, we normalize the PCA space (the PCs of each cell are normalized to unit norm using sklearn.preprocessing.normalize() and accordingly adjust the smoothing parameter.

**Cluster-purity score.** We define a cluster-purity score as a measure to distinguish between indicative and non-indicative clusters with respect to a given label (here we consider the cell type). Formally, for a given cluster of size $n$, we find the most frequent label, $y$, with $m$ cells. Now, we say that a cluster is indicative if, $\frac{m}{n} > \tau$, for a given threshold value $\tau$. This measure quantifies the homogeneity of the cluster.

**Statistics and reproducibility**

SiFT was evaluated across six publicly available datasets, using as many samples as possible in these datasets (no statistical method was used to predetermine the sample size). Datasets include drosophila wing disc development atlas (sample size = 27,016), human cell atlas (sample size = 486,134), virtual tumor dataset based on mouse embryonic stem cells (sample size = 279), mammalian liver data (sample size = 11,491), liver regeneration mouse spatial transcriptomics data (sample size = 14,798), and a COVID-19 dataset including PBMC samples (sample size = 624,322). Preprocessing steps, including quality control, were performed according to standard practice and reported for each dataset independently. The experiments involved running computational methods on previously published, publicly available datasets and did not require randomization. The investigators were not blinded to allocation during experiments and assessment of outcome.

**Reporting summary**

Further information on research design is available in the Nature Portfolio Reporting Summary linked to this article.

## Data availability

All datasets used in this study are publicly available. All relevant data supporting the key findings of this study are available within the article and its Supplementary Information files. The processed drosophila wing development data used in this study can be downloaded from https://figshare.com/articles/dataset/scvi-tools-reproducibility_processed_data/14374574/1?file=27458846. The processed Heart Cell Atlas used in this study can be found at https://www.heartcellatlas.org/v1.html; under visualisations/Downloads/Heart Global choosing the H5AD version (or directly downloaded by pressing the link: https://cellgeni.cog.sanger.ac.uk/heartcellatlas/data/global_raw.h5ad) The virtual tumor dataset used in this study can be downloaded from https://github.com/Kchen-lab/Cyclum/tree/master/old-version/data/mESC. The Mammalian liver data used in this study is available in the GEO database under accession code GSE145197. The liver regeneration data is available in the GEO database under accession code GSE223560. The processed COVID-19 dataset can be downloaded from https://www.covid19cellatlas.org/index.patient.html#publication under Datasets/Haniffa lab by choosing the H5AD version (or directly downloaded by pressing the link: https://covid19.cog.sanger.ac.uk/submissions/release1/haniffa21.processed.h5ad. The code to reproduce the results using the above datasets is available at https://github.com/nitzanlab/sift-sc-reproducibility. Source data are provided with this paper.

## Code availability

Software is available at https://github.com/nitzanlab/sift-sc and documentation at https://sift-sc.readthedocs.io[56]. The code to reproduce the results is available at https://github.com/nitzanlab/sift-sc-reproducibility.

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

## Acknowledgements

We thank Michal Klein for his thoughtful review and assistance in the software development and Klaas Mulder and Sabine Tanis, for ongoing discussions regarding the presented framework. We would further like to thank Kristen Koenig and Inbal Avraham-Davidi for stimulating biological discussions and Aleksandrina Goeva for helpful insight. We acknowledge Noa Moriel, Jonathan Karin, and all members of the Nitzan lab for general feedback. This work was funded by a scholarship for

outstanding doctoral students in data science, by the Israeli Council for Higher Education, and the Clore Scholarship for Ph.D students (Z.P.), an Azrieli Foundation Early Career Faculty Fellowship, the Israel Science Foundation (Grant no. 1079/21), and the European Union (ERC, DecodeSC, 101040660) (M.N.). Views and opinions expressed are however those of the author(s) only and do not necessarily reflect those of the European Union or the European Research Council. Neither the European Union nor the granting authority can be held responsible for them.

## Author contributions

Z.P. and M.N. conceived the study and designed the research; Z.P. implemented the method and analyzed the data, with guidance from M.N.; Z.P. and M.N. wrote the paper.

## Competing interests

The authors declare no competing interests.
