## [Peer Review File · Nature Communications]

SiFT: Uncovering hidden biological processes by probabilistic filtering of single-cell dataEditorial Note: This manuscript has been previously reviewed at another journal that is not operating a transparent peer review scheme. This document only contains reviewer comments and rebuttal letters for versions considered at *Nature Communications*.

Reviewer #5 (Remarks to the Author):

I believe the question raised by Reviewer 2 is satisfactorily addressed.

Most questions raised by Reviewer 4 are also satisfactorily addressed, namely major comment 1, and minor comments 1, 2, and 5. Specifically, regarding major comment 1, I believe that the authors illustrated in this version an innovative way to analyze scRNA-seq data. The results are interesting and likely appealing to readers. However, minor comments 3 and 4 could be addressed more thoroughly.

Minor comment 4 mentioned, and I concur, that "quadratic time complexity" is a serious problem. Thus, the author should explicitly mention the "quadratic time complexity", and perhaps possible algorithmic tricks to mitigate it. Only mentioning that a given number of cells can be handled by a GPU is not sufficient. Toning down the claim, as the original comment recommended, in the abstract is also a good idea. I was not expecting quadratic complexity and the GPU requirement when I saw "naturally scale". In addition, the space complexity for K is also quadratic. For the Heart Cell Atlas data with 500,000 cells, this means 250G entries. I assume is addressed by the "row-wise" computation approach mentioned in the response letter, but implementation details like this that are critical to scalability, should be clearly explained in the article to support the claim.

For minor comment 3, I am not sure the scib metrics are comprehensive enough on judging the method. Specifically, corrected data are rarely used for statistical tests because the distorted distribution can lead to unreliable results. Instead, it is recommended to run tests that take covariates into consideration on raw data (Luecken 2019 Molecular Systems Biology). While this study shows that breaking this rule can lead to interesting findings, the readers should be advised that DE analyses are not systematically examined for it and the p-values need to be interpreted with caution. In fact, I would appreciate a more systematic examination/calibration of commonly used DE analysis methods on "SiFTed" data, but I can understand if the authors consider it to be out of the scope of this article.

Response to reviewers comments for manuscript NCOMMS-23-50986-T

We thank the reviewer for their constructive feedback for our manuscript. We have addressed all remaining comments and revised the manuscript accordingly. We provide a point-by-point response below.

Point-by-point response to Reviewer #5 comments

Remarks to the Author

I believe the question raised by Reviewer 2 is satisfactorily addressed.

Most questions raised by Reviewer 4 are also satisfactorily addressed, namely major comment 1, and minor comments 1, 2, and 5. Specifically, regarding major comment 1, I believe that the authors illustrated in this version an innovative way to analyze scRNA-seq data. The results are interesting and likely appealing to readers. However, minor comments 3 and 4 could be addressed more thoroughly.

Comments

1. *Minor comment 4 mentioned, and I concur, that “quadratic time complexity” is a serious problem. Thus, the author should explicitly mention the “quadratic time complexity”, and perhaps possible algorithmic tricks to mitigate it. Only mentioning that a given number of cells can be handled by a GPU is not sufficient. Toning down the claim, as the original comment recommended, in the abstract is also a good idea. I was not expecting quadratic complexity and the GPU requirement when I saw “naturally scale”. In addition, the space complexity for K is also quadratic. For the Heart Cell Atlas data with 500,000 cells, this means 250G entries. I assume is addressed by the “row-wise” computation approach mentioned in the response letter, but implementation details like this that are critical to scalability, should be clearly explained in the article to support the claim.*

A: We thank the reviewer for this comment. To address this point we have toned down the “naturally scale” statement from the abstract and verified that in every reference to scalability in the main text the dependence on GPU support is mentioned. In addition, we have revised the methods section “runtime considerations” to include implementation details concerning the framework’s scalability to support our claims. Specifically we relate to the following:

- a. row-wise batching; as GPU memory is limited, similarly to the notion of batches in training neural networks we implemented a row-wise batching mechanism in SiFT. Importantly, this is naturally incorporated in the implementation and does not require any user engagement. Algorithmically this is sensible as by definition the normalization in SiFT is only required over rows of the kernel.

- b. sparse matrices support; In addition, whenever possible we perform computation over sparse matrices. This may be the case, for example, if the input “count matrix” is sparse and a *knn kernel* is computed over it. Again, this is the default performance of SiFT and does not require any additional input from the user.
2. *For minor comment 3, I am not sure the scib metrics are comprehensive enough on judging the method. Specifically, corrected data are rarely used for statistical tests because the distorted distribution can lead to unreliable results. Instead, it is recommended to run tests that take covariates into consideration on raw data (Luecken 2019 Molecular Systems Biology). While this study shows that breaking this rule can lead to interesting findings, the readers should be advised that DE analyses are not systematically examined for it and the p-values need to be interpreted with caution. In fact, I would appreciate a more systematic examination/calibration of commonly used DE analysis methods on “SiFTed” data, but I can understand if the authors consider it to be out of the scope of this article.*

A: We agree with the disclaimer the reviewer raises regarding DE tests. However, in general, there still isn't a consensus regarding *optimal* DE analysis for single-cell data and furthermore the robustness across datasets for DE tools is low [1,2,3]. In addition, given the increase in complexity of single-cell datasets and ongoing attempts to integrate large scale atlases across samples, dedicated papers have addressed a more specific question - DE analysis over integrated data, which is related to our setting as the discussion requires addressing data correction. For example Nguyen et al. [4] compared various workflows for DE analysis of scRNA-seq data with multiple batches in diverse settings including post batch effect correction. However, the complexity of the setting limits the ability to derive a “rule of thumb” applicable to all cases. Hence, in SiFT we took an explorative approach which, as stated by the reviewer, showed that DE analysis over corrected data can lead to interesting biological findings. Thus, while performing a deeper examination of this is intriguing, we indeed, as the reviewer notes, find it beyond the scope of this article, and an interesting direction for future work, as we now note in the revised discussion.

- [1] Heumos, L., Schaar, A.C., Lance, C. et al. Best practices for single-cell analysis across modalities. *Nat Rev Genet* (2023). <https://doi.org/10.1038/s41576-023-00586-w>
- [2] Samarendra Das, Anil Rai, Michael L Merchant, Matthew C Cave, and Shesh N Rai. A comprehensive survey of statistical approaches for differential expression analysis in single-cell term`RNA` sequencing studies. *Genes (Basel)*, 12(12):1947, December 2021.
- [3] Tianyu Wang, Boyang Li, Craig E. Nelson, and Sheida Nabavi. Comparative analysis of differential gene expression analysis tools for single-cell term`rna` sequencing data. *BMC Bioinformatics*, 20(1):40, Jan 2019. URL: <https://doi.org/10.1186/s12859-019-2599-6>, doi:10.1186/s12859-019-2599-6
- [4] Nguyen, H.C.T., Baik, B., Yoon, S. et al. Benchmarking integration of single-cell differential expression. *Nat Commun* 14, 1570 (2023). <https://doi.org/10.1038/s41467-023-37126-3>

Reviewer #5 (Remarks to the Author):

The authors satisfactorily addressed all my concerns. I recommend this one for publication.